# A Two-Class Data Transmission Method Using a Lightweight Blockchain Structure for Secure Smart Dust IoT Environments

**DOI:** 10.3390/s20216078

**Published:** 2020-10-26

**Authors:** Joonsuu Park, KeeHyun Park

**Affiliations:** Department of Computer Engineering, Keimyung University, Daegu 42601, Korea; parkjoonsuu@gmail.com

**Keywords:** internet of things, blockchain, lightweight, security, smart dust, urgent data, data integration

## Abstract

In smart dust IoT environments, a large number of devices with low computing power/resources are deployed to collect surrounding information. There are many issues to consider for an efficient and secure smart dust IoT environment. Sometimes the urgent sensed data needs to be transmitted immediately. In addition, there are potential problems related to security issues since the smart dust IoT systems may be deployed in hard-to-access areas. In this paper, we propose an effective transmission method for two-class sensed data for secure smart IoT systems. We divide the sensed data into two classes which consist of the urgent sensed data class (requiring urgent data transmission) and the normal sensed data class (with a slight transmission delay due to yielding to the urgent data transmission). In addition, for security reasons, the proposed transmission method uses two kinds of blockchains with the following two ledgers: (1) the urgent sensed data ledger, which is a ledger of data that needs urgent transmission; and (2) the normal sensed data ledger, which is a ledger of data that allows some delay. To be specific, the lightweight blockchain based on our earlier work is used for the normal sensed data transmission, whereas the modified conventional blockchain is used for the normal sensed data transmission. The experiments show that the performance of the proposed transmission method is better than the conventional transmission method in almost all sections. There is a 53% performance increase on average with regard to the transmission time. When the ratio of urgent sensed data is 0% (i.e., no urgent sensed data at all), the proposed transmission method is greater improved by as much as about 96%. This means that the lightweight blockchain scheme used in the proposed transmission method for the normal sensed data is very efficient.

## 1. Introduction

The IoT (Internet of Things) is a concept that includes the close connection between objects and people around them are established on a network [1,2,3,4,5,6,7,8]. The concept is already being used in many fields around us (e.g., transportation, smart city, smart home, electronic governance, electronic education, personal health, retail, logistics, agriculture, manufacturing industry, business, etc.) [9,10,11], and even many of the latest electronic products are equipped with IoT functions by default.

A smart dust IoT system is one of the IoT areas that monitors physical spaces (e.g., buildings, roads, clothing, and bodies) and deploys very small-sized sensors (i.e., smart dust IoT devices) around them to collect information such as temperature, humidity, acceleration, and pressure [2,5,6,7,8,9,10,11,12,13]. The very small size of smart dust IoT devices enables ultra-large deployment. In particular, these advantages make it possible to deploy devices using aerial spraying means in hard-to-reach areas. Additionally, smart dust devices that must be distributed in very large quantities are inevitably required to be manufactured at a low cost.

Some of the data collected in smart dust IoT environments require very urgent transmissions, while other data allow some delays. For example, among the data of personal healthcare devices, SpO_2_ (Saturation of percutaneous Oxygen) data can cause serious problems when the data transmitted experiences delay, but blood sugar data does not. Among climate data, seismic data can cause serious problems if not reported immediately, but some delays can be tolerated for humidity data transmission.

In our earlier study [14], we proposed a two-phase data reduction method that reduces the bandwidth in smart dust IoT environments. We classified data into normal data which can be delayed slightly, and urgent data which does not allow delay. Urgent data must be sent as soon as it is sensed. The study reduced the total size of the data to be transmitted as well as the network resources to be required. However, there are still potential problems related to security issues with smart dust IoT environments.

Smart dust IoT systems in hard-to-reach areas are easily exposed to various security threats such as data integration and authentication attacks much more than the conventional IoT systems. The blockchain method is generally known as a good solution for these attacks [15,16,17,18,19,20,21,22,23,24,25,26]. Unfortunately, however, smart dust IoT devices can neither store the ledger of the blockchain nor handle the mempool tasks because of limited computing power/resources. Moreover, in the blockchain, as the number of nodes increases, the processing speed of transactions decreases dramatically [16,17,18,23,25]. Therefore, in our earlier study [19], the concept of a lightweight blockchain was proposed for secure smart IoT systems with very limited computing capabilities. The study removed the blockchain functions that are not needed in the smart dust IoT environment and changed the blockchain structure to reduce the time required for smart dust device authentication. However, all the sensed data were treated homogeneously. That is, all the sensed data were transmitted in the same way and urgent data could not be treated appropriately. In this paper, we propose a two-class data transmission method using two types of blockchains (i.e., a lightweight blockchain and the conventional blockchain) to ensure the security of the smart dust IoT environment and to transmit the urgent data smoothly.

Therefore, we propose in this paper an effective transmission method for two-class sensed data for secure smart IoT systems. We divide the sensed data into two classes which consist of the urgent sensed data class (requiring urgent data transmission) and the normal sensed data class (with a slight transmission delay due to yielding to the urgent data transmission). In addition, for security reasons, the proposed transmission method uses two kinds of blockchains with the following two ledgers: (1) the urgent sensed data ledger, which is a ledger of data that needs urgent transmission; and (2) the normal sensed data ledger, which is a ledger of data that allows some delay. To be specific, the lightweight blockchain based on our earlier work [19] is used for the normal sensed data transmission, whereas the modified conventional blockchain is used for the normal sensed data transmission. Urgent sensed data is transmitted to the smart dust IoT server instantly before being managed as a block, while normal sensed data is accumulated and recorded in the normal data ledger for later transmission. The proposed method extracts the normal sensed data from the normal sensed data ledger and integrates/compresses the data in order to reduce the size of data transmission.

The remainder of this paper is organized as follows: Section 2 explains the smart dust IoT environments, the blockchain, the two-phase data reduction method, and the lightweight blockchain. In Section 3, we propose the two-class data transmission method for a secure smart dust IoT system with the lightweight blockchain. Section 4 shows the experimental results of the proposed method which demonstrates a significant reduction in data transmission time. Finally, Section 5 discusses the conclusions.

## 2. Related Works

### 2.1. Smart Dust IoT Environments

The smart dust technology integrates various technologies for microscopic sensor devices [2,3,4,5,6,7,8,9,10,11,13]. As the name suggests, smart dust IoT devices not only refer to devices of small size like dust but also include their IoT communication function [9]. Smart dust IoT devices are not only very small but also deployable in very large numbers, and are mainly deployed in hard-to-reach areas using airborne machines. Therefore, a smart dust IoT device must inevitably be a low-cost device with very limited computing power/resources. The characteristics of the smart dust IoT devices bring many challenges to smart dust IoT systems.

In our earlier study [8], a layered smart dust IoT system was proposed in order to alleviate the bottleneck phenomena and to predict dynamically the appropriate plane partitions for given workloads. Figure 1 shows an overview of physical devices in a smart dust IoT environment [8]. As shown in Figure 1, the smart dust IoT system consists of three layers. SDDs (Smart Dust Devices) are sensors gathering data around them. RDDs (Relay Dust Devices) are SDDs with enough computing power/resources to process data. The processing node in the smart IoT server is a processor that processes the data received from RDDs, and the PC (Pool Control) node is a processor node that distributes processing loads efficiently to the multiple processing nodes according to the proposed dynamic partition algorithm.

### 2.2. The Blockchain

The blockchain is a secure distributed computing technology that processes all actions as transactions, chains the blocks that stored transactions, and distributes blocks to multiple nodes in the network [22]. Distributed ledgers are considered a solution to problems such as data tampering, tampering, and double payments since no one can arbitrarily make modifications and anyone can view the change results [21,22]. In broad terms, the blockchain is a system or algorithm that stores many transactions (including details of authentication information, details of changelog, etc.) in blocks, and chains the blocks to the chain using a hash [16,23,27,28,29].

In [17], the header of each block has the hash information of the preceding block. This causes a chain effect if an attacker wants to forge the information of a block, and then he/she must modify the node repeatedly in front of it, and again in front of it. That is, in order to forge one piece of information, the attacker must change the information of all the blocks in the blockchain which is distributed across the network. Since the amount of computation increases in real-time as new blocks and transactions are constantly generated, from the moment the number of blocks/transactions exceeds the threshold, the amount of computation that increases in real-time becomes greater than the amount of computation performed. In this way, the integrity of the blockchain is always guaranteed [17,18,19,20,21]. However, as the number of participating nodes increases, the number of blocks increases, and hence the blockchain starts to slow down at a tremendous rate [16].

### 2.3. A Lightweight Blockchain for a Smart Dust IoT System

N.H. Kim, S.M. Kang, and C.S. Hong focused on the basis that the block size of the blockchain continues to increase [30]. The authors said that the fees increased because the size of the ledger increases as the number of nodes in the network increases, which is the reason why service providers cannot use the blockchain for micropayments. In the study, the service provider owns the entire ledger, and each group owns only the ledger that the group needs. That is, service providers constitute a large blockchain, and each group has a small, lightweight blockchain. When a transaction occurs, the proposed system updates the ledger of the service provider and the ledger to which the group includes the device that generated the transaction belongs. The system does not update transactions made by devices belonging to other groups by configuring the lightweight blockchain, thereby significantly reducing transaction costs. However, the system may experience a problem if the number of devices in the group decreases, that system could not be trusted to perform the verification procedure of the blockchain.

A. Dorri, S.S. Kanhere, R. Jurdak, and P. Gauravaram proposed a method of storing and managing ledgers in pre-authenticated local storage [31]. The authors focused on the basis that transactions occurring in an already authenticated blockchain could verify data integrity. The blockchain system proposed in [31] is very light and operates reliably. However, the system is very low in utilizations because it builds an ecosystem isolated from the outside. Therefore, the authors designed a layered system to combine external ecosystems and their inherent ecosystems. They designed the system so that the types of transactions could be divided into local transactions and overlay transactions so that they could be processed differently. The studies in [31] focused their attention on device authentication for IoT. The system effectively certifies devices to meet their design goals. However, if a large amount of data is generated (like in a smart dust IoT environment), there would be numerous conversion exchanges between the overlay network and the local network, and the efficiency of the system decreases rapidly.

Smart dust IoT environments have potential security issues such as data forgery attacks by unauthorized devices. The blockchain attracts much attention as a solution to the security problems for conventional networks. However, the conventional blockchain cannot operate smoothly in the smart dust IoT system where most of the SDDs have very limited computing power/resources. Therefore, in our earlier study [19], we proposed a lightweight blockchain that can be applied to smart dust IoT environments. The underlying smart dust IoT system used in [19] is similar to the system in [8] except that the Auth node and the Time node are added to the system in [8]. The system [19] is designed in a lightweight fashion in order to be operated sufficiently in smart dust IoT devices with very limited computing power/resources. The concepts of some conventional characteristics (i.e., blockchain mining and mempool, etc.) are excluded in the system. However, in order to solve the synchronization problem caused by removing various functions of the blockchain, the system introduces the concept of standby ledgers and the concept of the Time Node (see Section 3 for detailed explanations on standby ledger, Time Node, and commit). The system efficiently solves the problems of smart dust IoT environments by using a lightweight blockchain. However, since the lightweight blockchain is not support equipped with a solution for transmitting urgent data considered in this paper, urgent data may not be transmitted in time or omitted, which may cause serious problems.

### 2.4. A Blockchain-Based Platform for Healthcare Information Exchange

S. Jiang, et al., proposed BlockHIE [32], an excellent blockchain-based platform for exchanging data of medical records and personal health devices. We believe that the study was one of the earliest studies to use two types of blockchains. They proposed two types of blockchains, EMR-Chain for medical records and PHD-Chain for personal health devices. They analyzed various requirements to share medical data from various sources and used two types of blockchain to process different types of medical data based on the analysis. The differences between our study and BlockHIE [32] are that we used two different types of blockchains as well as different processing algorithms. Specifically, we used the conventional blockchain for the urgent data, and the lightweight blockchain for the normal data.

### 2.5. Two-Phase Data Reduction Method

In environments where a massive amount of data occurs explosively, such as Smart dust IoT environments, the network bandwidth of the system is insufficient in most situations. Focusing on the fact that normal data can be “delay-tolerant” in the first phase of the two-phase data reduction method in our earlier study [14], the normal sensed data is transmitted with a slight delay in order to supply the surplus network bandwidth to the urgent sensed data transmission in such a way that the urgent sensed data can be transmitted immediately. Additionally, in the first phase, the amount of data transmissions is reduced by not sending the normal sensed data if the difference between the previous sensed data value and the current sensed data value does not exceed the threshold value. In the second phase, the size of the transmitted data is reduced by sharing the network information (e.g., protocol header, etc.).

Even though the method supports the smooth transmission of urgent data from the smart dust IoT system, it cannot solve problems such as authentication and data integrity issues.

## 3. A Two-Class Data Transmission Method Using a Lightweight Blockchain Structure

The key considerations in this study are as follows:Immediate transmission of the urgent sensed data by classifying two-class dataEfficient sensed data transmission for a secure smart IoT system with very limited computing power/resources by using the lightweight blockchain

The proposed system in this study is based on the layered smart dust IoT system proposed in our earlier studies [8,17]. Figure 2 shows the physical device configuration of the smart dust IoT system using the lightweight blockchain. The two nodes in the figure need to be mentioned. The Auth node sends authentication information to RDDs. The Time node plays the role of determining synchronization time. A detailed description of the two nodes can be found in [19]).

Both the authentication information and the sensed data transmissions are handled as transactions. The proposed system applies a ledger-based data transmission method to reduce the vast amount of data generated in smart dust IoT environments. That is, instead of transmitting all the sensed data collected in each RDD to the smart dust IoT server, only the selected specific RDD parses all the ledger information propagated from every RDD in order to transmit the smart dust IoT server (see Section 3.1 for details). When data is collected and transmitted, transactions occur. The authentication and registration information of every SDD are also treated as transactions.

When the blockchain is applied to the smart dust IoT system and processing the ledger, there may be some delays while urgent sensed data is being processed by the ledger. In order to solve the problem, we decide to raise the concept of the mempool to the ledger level on the conventional blockchain (e.g., Ethereum). That is, we created two kinds of ledgers that manage two-class data (urgent sensed data and normal sensed data) to treat them separately, which means that there are two blockchains (i.e., lightweight blockchain and the (modified) conventional blockchain) in the system. We describe the normal data and the urgent data processing procedures in Section 3.1 and Section 3.2, separately.

### 3.1. The Normal Sensed Data Processing Procedure

The normal sensed data is not urgent and hence does not require immediate transmission if the network bandwidth is insufficient. RDDs collect, integrate, and then transmit the normal sensed data. While the normal sensed data is being processed, network resources are essentially free, so the urgent sensed data can be transferred immediately. The lightweight blockchain proposed in this study is used for authentication and data integrity.

Table 1 shows the normal sensed data blockchain processing procedure.

Figure 3 shows the sequence diagram for the normal sensed data blockchain processing procedure mentioned above. Our earlier work [19] did not consider data integration before transmission. In this study, the sequence diagram was updated by adding the data integration transmission stage in the box at the bottom of Figure 3.

#### 3.1.1. Creating and Propagating a New Block/Transaction

Transactions in the conventional blockchain can be identified uniquely by such information as the hashes of the transactions, mempool [16,22,23,24,25,26], etc., regardless of the order of transactions. However, maintaining the conventional blockchain in the smart dust IoT devices is a very difficult task because the devices have very limited computing power/resources. The proposed lightweight blockchain for the smart dust IoT system cannot uniquely identify transactions. To solve the problem, we use the order of transactions as a unique identification tool for transactions. We introduce the standby ledger, commit, and the Time node to ensure the order of transactions.

A standby ledger is a temporary ledger of the blocks not registered in the normal data ledger. Blocks/transactions are registered temporarily in the standby ledger until they are committed. The commit is to merge the normal data ledger and the standby ledger at the commit time, which is the time determined by the Time node. The Time node performs scheduling using the scheduling table to determine the commit time. The scheduling table is a very simple table that is divided by time intervals long enough to commit the standby ledger to the normal data ledger. In summary, the following two facts can guarantee that the transactions are out of order: (1) the commit time must be allocated before the transaction can be recorded in the standby ledger, (2) the Time node does not allocate a commit time overlapped with other commit times.

The conventional blockchain operates by owning blocks through mining and rewarding hash operations by fees. However, the system in this study should be able to be used even in ecosystems that do not generate rewards (fees), and hence the concept of mining is excluded in this study. Instead, we focus on the nature of the blockchain, where blocks are used as a tool for chaining, and also focus on another characteristic of the blockchain wherein transactions are treated as details. This is the same approach for the urgent sensed data blockchain, which will be described later.

The creating and propagating of a new block/transaction stage creates a block/transaction and puts a transaction into a block. The stage begins by creating a transaction. After the transaction is created, the RDD’s device’s information is recorded in the “to” field, the SDD’s device’s information that collected the data is recorded in the “from” field, and the collected information is recorded in the “data” field. (Figure 4 shows an example of the blockchain used in a normal sensed data blockchain for this study.) After that, the device’s information and the hash of the previous block are written in the block, and RDD requests a commit time to the Time node. The time node that has received a request selects the earliest time in its scheduling table as the commit time. The selected commit time is marked in the scheduling table so that it cannot be allocated to other commit times. After the RDD receives the commit time from the Time node, the RDD records the commit time in the “Sync Time” field of the block. Finally, the block/transaction creation stage is completed by writing the result of hashing the current block by SHA 256 in the hash field.

#### 3.1.2. Registration in the Standby Ledger

After the block is created, it is propagated to all other RDDs, which is similar to the conventional blockchain. The only difference is that a transaction is given to the mempool in the normal blockchain, while a transaction is recorded in standby ledgers in the proposed lightweight blockchain.

#### 3.1.3. Synchronizing and Integrating Data Transmission

In the synchronization and integrated data transmission stage, by committing the blocks of the standby ledger to the normal data ledger, synchronization is performed and the collected data is transmitted to the smart dust IoT server. The synchronization step simply appends the standby ledger to the normal data ledger. This stage includes parsing the data in the normal data ledger and integrating the parsed data.

The synchronization and integrated data transmission stage consist of three main parts: synchronization, extraction, and integrated transmission. Synchronization is the simple task of committing the transactions in the standby ledger to the normal data ledger. Extraction parses/extracts data from the standby ledger. The standby ledger has the transactions created from the previous commit to the present. Finally, the integrated transmission integrates and transmits the data parsed/extracted from the standby ledger. The method used for the integration work is the two-phase data reduction method proposed in our earlier study [14]. In the threshold filtering phase, which is the first phase of the two-phase data reduction method, only data that shows a difference between the previous sensed data and the current sensed data that is above the threshold is transmitted. The integration phase, the second phase of the two-phase data reduction method, integrates duplicated information (e.g., SDD’s ID, the header of the communication protocol, etc.) in each data packet. Therefore, in the integration phase, the size of transmitted data is reduced. Table 2 shows the detailed procedure of the synchronization and integration transfer stage.

### 3.2. The Urgent Sensed Data Processing Procedure

The normal sensed data can be delayed for transmission, whereas the urgent sensed data cannot. As a result, the standby ledger, Time node, commit operation, and the two-phase data reduction method cannot be applied to the urgent sensed data blockchain. Therefore, we manage the urgent sensed data blockchain through the urgent data ledger. Table 3 shows the urgent sensed data blockchain processing procedure.

When an SDD transmits the sensed data to a neighboring RDD, the RDD identifies that the data is urgent by viewing the header of the data. Since the transmission of urgent sensed data cannot be delayed, the RDD immediately transmits the data to the smart dust IoT server and then starts creating a new block and a new transaction. The process of creating a transaction and a block is almost the same as the process for the normal sensed data. The only difference is that after all the information of the transaction is stored, the result of hashing the transaction through SHA 256 is recorded in the ‘hash’ field of the transaction. The hash of the transaction allows the user to distinguish each transaction even in a mixed state regardless of the transaction order. The block is then propagated immediately to other blocks. On receiving the propagated block, every RDD updates its urgent sensed data ledger for data integrity. Figure 5 shows the sequence diagram of the urgent sensed data processing procedure. The urgent sensed data is transmitted as soon as it arrives at the RDD and is managed as a block after transmission.

Figure 6 shows an example of the urgent sensed data blockchain. Unlike the normal sensed data blockchain, the hash of the previous transaction, which is similar to the feature of the mempool, is used to uniquely identify the transaction.

Figure 4 shows the lightweight blockchain for the normal sensed data, and Figure 6 shows the blockchain for the urgent sensed data. The biggest difference between the two figures is that Figure 6 has a hash of the transaction. This means that transactions can be uniquely identified even if they are not synchronized. Conventional blockchain uses the mempool to perform transaction synchronization. However, considering the target of the lightweight blockchain (i.e., devices with very limited performance/resources), it is difficult to use the mempool.

A hash of a transaction allows the transaction to be uniquely identified but generates additional actions such as creating a hash and detecting the transaction to synchronize the order. The hash of a transaction allows the transaction to be uniquely identified but also causes additional actions such as creating and detecting a hash. Therefore, a transaction hash is not used for the normal sensed data that does not necessarily need a transaction hash (see Figure 3), while a transaction hash is used for the urgent sensed data that requires a transaction hash (see Figure 5).

## 4. Experiments

We conducted experiments to show that the proposed transmission method can reduce the size of the data transmission, alleviate the bottleneck phenomena, and can secure the network bandwidth for urgent sensed data. This experiment was simulated in the environment shown in Table 4.

The program modules developed for the experiments are shown in Figure 7.

This system is largely divided into four physical devices: SDDs, RDDs, the IoT Server, and the Time Node.

Among the software of the four physical devices, the common modules are as follows:Init Module: performs initialization according to the role of each deviceSession Module: manages the communication connection to the deviceSending Module: sends data generated or processed from a device to other devicesReceiving Module: receives data from other devices

The SDD software consists of the following modules:Sensing Module: senses/collects data in a neighboring area

The RDD software consists of the following modules:BlockMng Module: creates and manages blocksTransMng Module: records transactions such as authentication and data collection in blocksBroadcasting Module: propagates blocks to other RDDs

The Time Node software consists of the following modules:TaskRegister Module: registers tasks in a scheduleTimeCheck Module: determines synchronization timeNotification Module: notifies when the synchronization time has reached

We performed experiments by changing the ratio of urgent sensed data to determine how efficient the proposed method is compared with the conventional method by which normal sensed data is transmitted immediately like urgent sensed data. The total number of the sensed data points used in each experiment case was 10,000. When the urgent sensed data ratio was 40%, we had 4000 urgent data points and 6000 normal data points. Each experiment case was administered 10 times to compute the average case result. Figure 8 shows the performance comparisons between the proposed transmission method and the conventional transmission method with blockchains, with regard to the transmission time. The data transmission time was measured from the time when the sensed data was generated in an SDD to the time when all the data were received by the IoT server.

As a result of the experiment, the performance of the proposed transmission method was better than the conventional transmission method in almost all sections. There was a 53% performance increase on average with regard to the transmission time. When the ratio of the urgent sensed data was 0% (i.e., no urgent sensed data at all), the proposed transmission method was greater improved by as much as about 96%. This meant that the lightweight blockchain scheme used in the proposed transmission method for the normal sensed data was very efficient.

Figure 9 shows the ratio of transmission data reduction when the ratio of urgent data changes. The rate of transmission data reduction decreased as the ratio of urgent sensed data increased, which indicated the same meaning as Figure 8.

One item of note is that the performance of the proposed transmission method was worsened (by about 1%) when the urgent data was 100%. This explained how much overhead the proposed method had in order to maintain the two types of blockchains. That is, the proposed method writes/manages two ledgers (the normal sensed data ledger and the urgent sensed data ledger), and hence the occupied memory was relatively bigger in the proposed method. It was found that the memory occupancy on average for the proposed method and the conventional method was about 79% and 43%, respectively.

Figure 10 shows the average waiting time for the urgent data as the data rate varies.

The results in Figure 10 show that the conventional transmission method provided more average waiting time by about 19% more time on average for the urgent sensed data. This meant that when the urgent sensed data was transmitted by the conventional transmission method, the urgent sensed data was delayed by 19%, compared to the proposed transmission method. In particular, when the ratio of the urgent sensed data was 10% (i.e., most of the transmitted data was the normal data), the average waiting time of the urgent data suffered a serious delay (by as much as 44%) in the conventional transmission method. In this case, the urgent data may not meet the urgent situation.

## 5. Conclusions

In this paper, we propose an effective transmission method for two-class sensed data for secure smart IoT systems. We divide the sensed data into two classes which consist of the urgent sensed data class (requiring urgent data transmission) and the normal sensed data class (with a slight transmission delay due to yielding to the urgent data transmission). In addition, for security reasons, the proposed transmission method uses two kinds of blockchains with the following two ledgers: (1) the urgent sensed data ledger, which is a ledger of data that needs urgent transmission; and (2) the normal sensed data ledger, which is a ledger of data that allows some delay. To be specific, the lightweight blockchain based on our earlier work [17] is used for the normal sensed data transmission, whereas the modified conventional blockchain is used for the normal sensed data transmission. Urgent sensed data is transmitted to the smart dust IoT server instantly before being managed as a block, while normal sensed data is accumulated and recorded in the normal data ledger for later transmission. The proposed method extracts the normal sensed data from the normal sensed data ledger and integrates/compresses the data in order to reduce the size of data transmission.

The experiments show that the performance of the proposed transmission method is better than the conventional transmission method in almost all sections. There is a 53% performance increase on average with regard to the transmission time. When the ratio of urgent sensed data is 0% (i.e., no urgent sensed data at all), the proposed transmission method is greater improved by as much as about 96%. This means that the lightweight blockchain scheme used in the proposed transmission method for the normal sensed data is very efficient.

In this paper, we considered data of two classes only, because more management work and overhead would be required as we consider a greater number of data classes. However, an elaborate data classification with little overhead may result in system performance improvement. Therefore, in the near future, we will prepare a study on a data processing/transmission algorithm with more than two data classes.

## Figures and Tables

**Figure 1 sensors-20-06078-f001:**
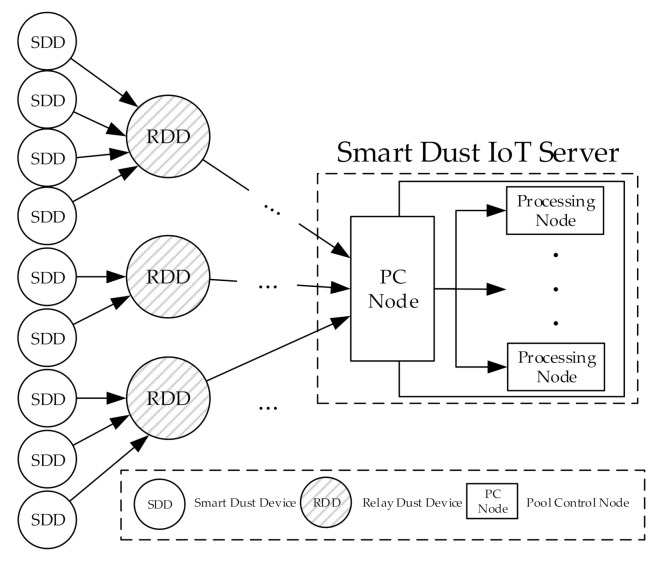
An overview of physical devices in a smart dust IoT environment [8].

**Figure 2 sensors-20-06078-f002:**
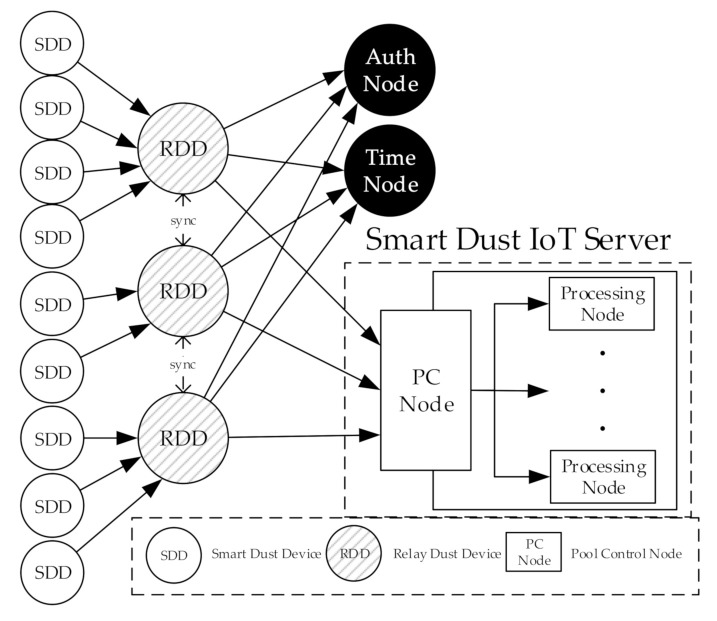
An overview of physical devices in a smart dust IoT environment with the lightweight blockchain.

**Figure 3 sensors-20-06078-f003:**
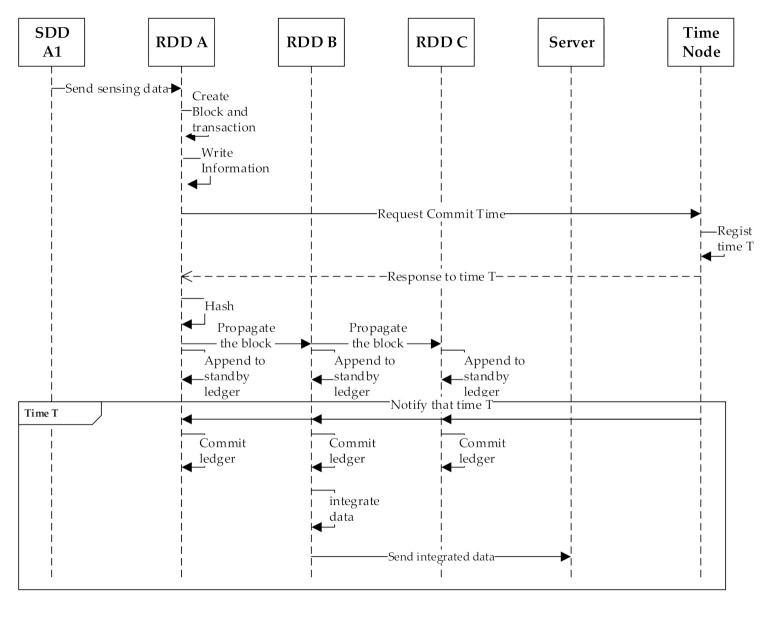
A sequence diagram for the normal data blockchain processing procedure.

**Figure 4 sensors-20-06078-f004:**
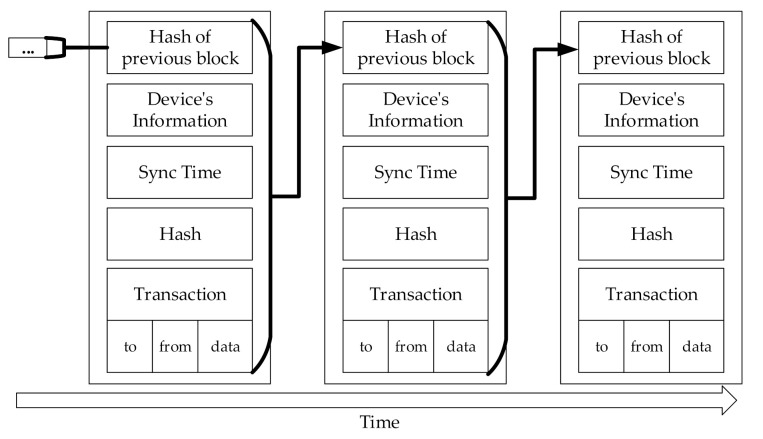
An example block for the normal sensed data blockchain.

**Figure 5 sensors-20-06078-f005:**
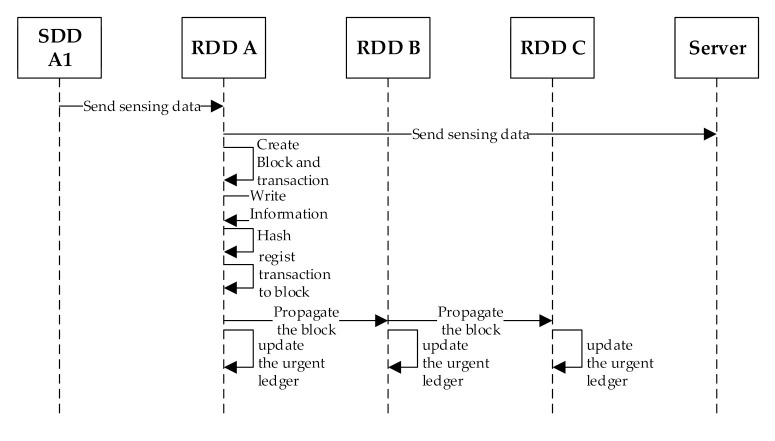
A sequence diagram for the urgent sensed data blockchain processing procedure.

**Figure 6 sensors-20-06078-f006:**
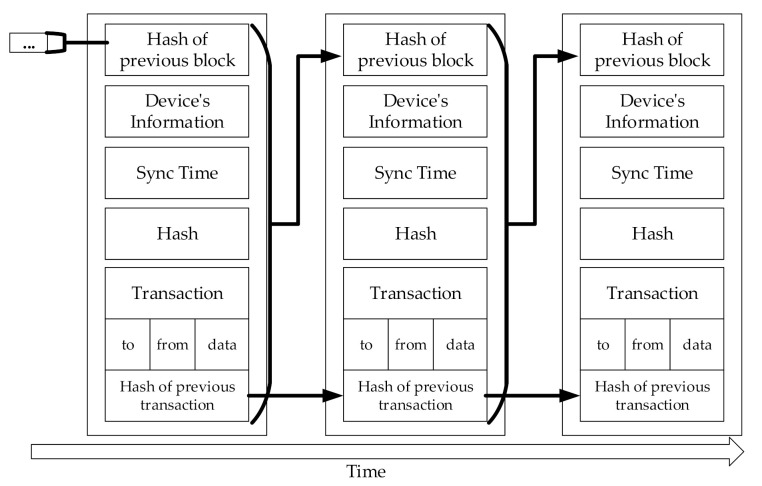
An example of the urgent sensed data blockchain.

**Figure 7 sensors-20-06078-f007:**
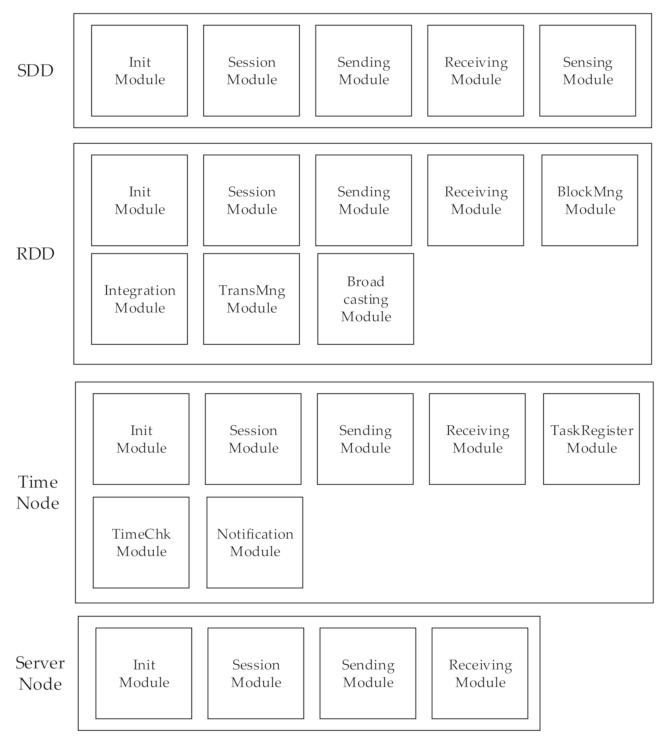
The software modules used in the experiment.

**Figure 8 sensors-20-06078-f008:**
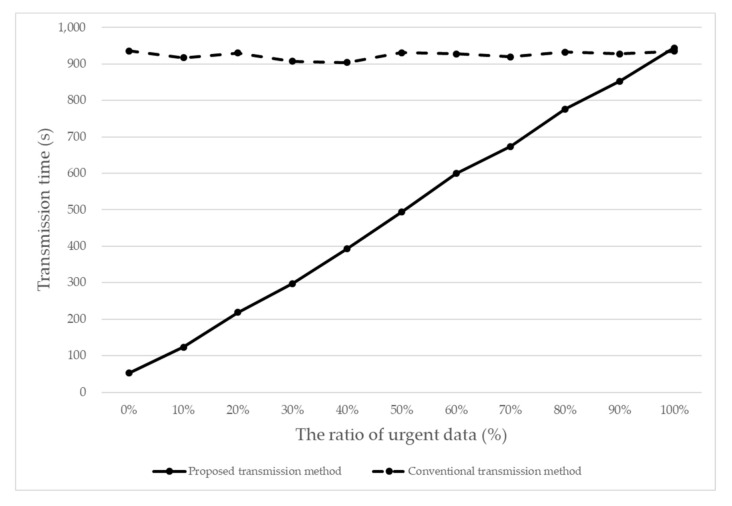
Performance comparisons between the proposed transmission method and the conventional transmission method.

**Figure 9 sensors-20-06078-f009:**
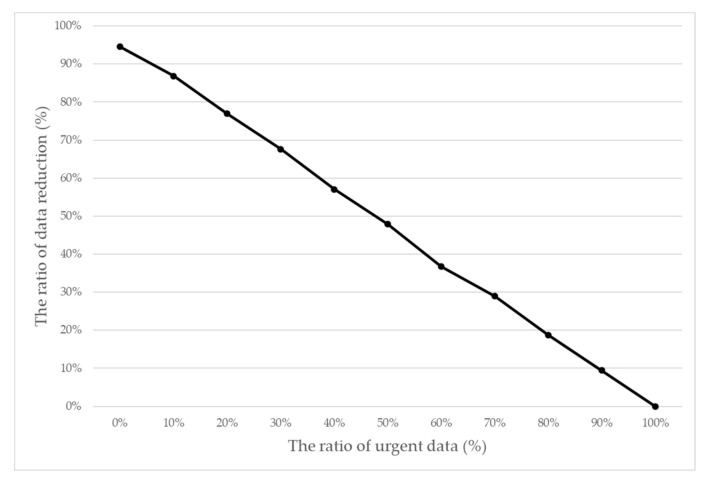
The ratio of transmission data reduction when the ratio of urgent data changes.

**Figure 10 sensors-20-06078-f010:**
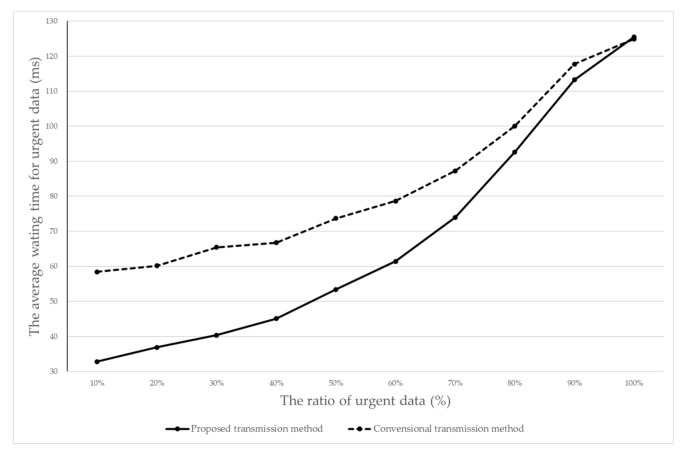
The average waiting time of the urgent data.

**Table 1 sensors-20-06078-t001:** The normal sensed data blockchain processing procedure.

Stages	Steps	Procedures
Generating data	1	An SDD transmits the normal sensed data to a neighboring RDD.
Creating and propagating a new block/transaction (see Section 3.1.1 for details)	2	The RDD received the normal data creates a new block and a transaction.
	3	The RDD writes the device (RDD and SDD) information and the normal sensed data to the Transaction and writes the transaction, the hash of the previous block, and the device (SDD) information to the block.
	4	The RDD requests the transaction commit time to the Time node.
	5	The Time node schedules the commit time, registers the time in its own scheduling table, and then returns the allocated commit time to the RDD.
	6	The RDD computes the hash of the block, writes it to the hash field of the block, and then propagates the block to other RDDs.
Registering the block in the standby ledger(see Section 3.1.2 for details)	7	All the RDDs (including the RDD that performed the block propagation) register the block in their respective standby ledgers.
Synchronizing the block and integrating data for transmission(see Section 3.1.3 for details)	8	The Time node commits the standby ledger to the normal data ledger when the commit time has been reached.
	9	The RDD, which added the last block to the standby ledger, extracts/compresses the normal sensed data from its ledger and then sends it to the smart dust IoT server using the two-phase data reduction method.

**Table 2 sensors-20-06078-t002:** A synchronization and integration transfer stage procedure.

Steps	Explanations
1	Synchronize by committing the standby ledger of each RDDs to the normal data ledger
2	Parse/extract the standby ledger data
3	Perform threshold filtering (the first phase of the two-phase data reduction method)
4	Perform integration filtering (the second phase of the two-phase data reduction method)

**Table 3 sensors-20-06078-t003:** The urgent sensed data blockchain processing procedure.

Stages	Steps	Procedures
Generating data	1	An SDD transmits the urgent sensed data to a neighboring RDD
Transferring data	2	The RDD transmits data to the smart dust IoT Server
Creating and propagating a new block/transaction	3	The RDD creates a new block and a new transaction
	4	The RDD writes the hash of the previous block and the device (SDD) information to the block, and the RDD writes the device (RDD/SDD) information and data to the transaction
	5	The RDD performs a hash operation on the transaction and performs an SHA 256 hash operation on the block
	6	Register the transaction in the block
	7	Propagating the block to other RDDs
Synchronizing the block	8	On receiving the propagated block, every RDD updates its urgent sensed data ledger

**Table 4 sensors-20-06078-t004:** Experiment environment.

Component	#1	#2	#3
CPU	Intel i7-6700	Intel i7-6700	Intel i9-9900
RAM	16GB	32GB	64GB
OS	Ubuntu 18.01 LTS	Ubuntu 18.01 LTS	Ubuntu 18.01 LTS

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
