# Peer review of "A Two-Class Data Transmission Method Using a Lightweight Blockchain Structure for Secure Smart Dust IoT Environments"

_sensors, 2020, doi:10.3390/s20216078_

Round 1

Reviewer 1 Report

Authors present a transmission method for two-class sensed data for secure smart IoT systems. The proposed transmission method uses two kinds of blockchains with following two ledgers - the urgent sensed data ledger, and  the normal sensed data ledger. Finally, they show that the performance of the proposed transmission method is better than the conventional transmission method.

The manuscript is well organized, the figures are clear and understandable, the conclusion supports the experiments.

Therefor I propose that the manuscript is accepted.

Author Response

Please see the the attachment.

Reviewer 2 Report

In this paper, the authors propose to divide smart dust IoT data into two classes based on their degrees of urgency and deal with the two classes of data using different blockchains. In the introduction, we see that smart dust IoT has different degrees of urgency indeed. The motivation is clear. The paper is easy to follow. Still, there are some comments as follows:
- It seems the division of the data into two classes, and the lightweight blockchain for transmitting data, are proposed in [14] and [19] and are not the contributions of this paper. Note that [14] and [19] are the earlier works by the authors. It raises my concern about whether the real contributions of this paper are enough. Actually, the authors should summarize the contributions in introduction for the evaluation.
- The abstract is too long and is not well written. Three challenges are mentioned in the first paragraph, but not all of them are the main challenges for this paper. For example, which algorithm or mechanism handles the challenge of large number of sensing devices?
- In abstract, "In this paper, we propose" will be better than "Therefore, we propose in this paper".
- The idea of dividing data into several classes and handle them in different blockchains is not new. For example, "BlocHIE: A BLOCkchain-Based Platform for Healthcare Information Exchange" also has a similar idea. The difference is that the paper targets for healthcare data.
- The part of performance evaluation is too weak. The author should enrich the experiments.

Author Response

We would like to express our sincere gratitude to the Editor-in-Chief and the reviewers for their insightful comments and suggestions. These comments have helped us to further improve the paper. We believe that the revised version addresses all comments from the reviewers.

(Reviewer #2)

  1. - It seems the division of the data into two classes, and the lightweight blockchain for transmitting data, are proposed in [14] and [19] and are not the contributions of this paper. Note that [14] and [19] are the earlier works by the authors. It raises my concern about whether the real contributions of this paper are enough. Actually, the authors should summarize the contributions in introduction for the evaluation.

Reply: Our previous study [19] proposed a lightweight blockchain that utilizes a tree structure. The paper [19] did not mention the urgent data transmission. The dividing of the data into two classes (urgent data/normal data) was studied in [14].

In fact, the paper [14] was presented at the MUE 2020 conference. And we were recommended by the MUE 2020 committee to submit the expanded version of the paper to the SENSOR special issue. That is why this paper refers to [14]. The paper [14] proposed a two-class data transmission method without considering security issues (no blockchains, either). In other words, this paper expands the paper [14] by using the blockchain concept to consider security issues. With the blockchain concept, most of the processing algorithms for the two-class data transmission in [14] are changed/added.

The major contribution of this paper is the use of two different types of blockchains for the two-class sensed data transmission for the secure smart dust IoT system. We added a statement about this to make our contribution more clear in this paper.

See: Lines 70 to 72, in Section 1 (page 2).

  1. - The abstract is too long and is not well written. Three challenges are mentioned in the first paragraph, but not all of them are the main challenges for this paper. For example, which algorithm or mechanism handles the challenge of large number of sensing devices?

Reply: Thank you. We have deleted some statements related to the bottleneck issue in the Abstract/Introduction section because this study focuses on such issues as security and the processing of the urgent sensed data. Also, we have deleted some other statements to make the abstract more tidy.

See: the first and the second paragraphs in the Abstract (page 1), the second paragraph in Section 1 (page 2).

  1. In abstract, "In this paper, we propose" will be better than "Therefore, we propose in this paper".

Reply: Thank you. We changed "Therefore, we propose in this paper" to "In this paper, we propose."

See: Line 16 in the Abstract (page 1)

  1. - The idea of dividing data into several classes and handle them in different blockchains is not new. For example, "BlocHIE: A BLOCkchain-Based Platform for Healthcare Information Exchange" also has a similar idea. The difference is that the paper targets for healthcare data.

Reply: Thank you for advising us about the related research. We have reviewed the paper titled “BlocHIE: A BLOCkchain-Based Platform for Healthcare Information Exchange.” We thought the idea of the paper was great and played a pioneering role in the dual blockchain mechanism. In fact, we have decided to introduce “BlocHIE: A BLOCkchain-Based Platform for Healthcare Information Exchange” as one of the related studies in Section 2.4 (reference [32]).

See: Lines 173 to 182 in Section 2.4 (page 4 to 5), Ref. no. 32 in the References (page 17).

  1. The part of performance evaluation is too weak. The author should enrich the experiments.

Reply: Thank you. We agree with your comment. We have thus performed an experiment on the average waiting time of the urgent sensed data and added the related graph/paragraphs in Section 4.

See: Figure 13 and the related statements from Lines 413 to 423 in Section 4 (pages 14 to 15).

Reviewer 3 Report

The article describes a data transmission method using a lightweight blockchain structure for secure smart dust IoT environments with two classes. 

The authors present a technique that seems to be used and verified in previous work. However, the new contributions are significant in quantitative and qualitative aspect in comparison with the previous work.

The following issues must be taken into account and solved properly.

Minor concerns:

  • Although the idea of the article is interesting it is not clear why the described method use only two classes;
  • Some information that are not in direct connection with the article and are well-known should be removed e.g. Fig.2 - An example of the conventional blockchain - a simple reference is enough
  • Fig 10 must be explained in more detail, as it is the main key of the article, vs. Fig.6 (An example block for the normal sensed data blockchain);
  • In the section of Conclusions, further improvements of the described method could give a plus for presentation of the article;
  • More references must be added by avoiding self-citations.

Major concern:

  • A more rigorous evaluation of the proposed method vs conventional method is absolutely necessary. Fig 13 seems to present the same information as Fig.12 but from other point of view. I think this is the main lack of the article.

Author Response

We would like to express our sincere gratitude to the Editor-in-Chief and the reviewers for their insightful comments and suggestions. These comments have helped us to further improve the paper. We believe that the revised version addresses all comments from the reviewers.

(Reviewer #3)

  1. Although the idea of the article is interesting it is not clear why the described method use only two classes;

Reply: We could consider more than two classes. In this case, we would have to prepare more management work than in the two-class scheme, which definitely creates more overhead.

   However, we thought the idea of having more than two classes is worth consideration for the future study. Therefore, we have added this as a future study in the Conclusion section.

See: Lines 444 to 448 in Section 5 (page 15).

  1. Some information that are not in direct connection with the article and are well-known should be removed e.g. Fig.2 - An example of the conventional blockchain - a simple reference is enough

Reply: We agree with your opinion. Fig. 2 in Section 2.2 was deleted along with some related statements. 

⇒See: Section 2.2 (pages 3 to 4).

  1. Fig 10 must be explained in more detail, as it is the main key of the article, vs. Fig.6 (An example block for the normal sensed data blockchain);

Reply: Thank you. We have added some additional explanatory statements for this comment, mostly on the differences between Fig. 5 and Fig. 9.

See: Lines 337 to 348 in Section 3 (page 11).

  1. In the section of Conclusions, further improvements of the described method could give a plus for presentation of the article;

More references must be added by avoiding self-citations.

Reply: Thank you. Your idea of having more than two classes is worth consideration for the future study. We have mentioned this in the Conclusion section.

⇒ See: Lines 444 to 448 in Section 5 (page 15).

  1. A more rigorous evaluation of the proposed method vs conventional method is absolutely necessary. Fig 13 seems to present the same information as Fig.12 but from other point of view. I think this is the main lack of the article.

Reply: Thank you. In order to provide the readers with a more comprehensive view, we performed another experiment on the average waiting time of the urgent sensed data and added the related graph/paragraphs in Section 4.

See: Figure 13 and the related statements from Lines 413 to 423 in Section 4 (pages 14 to 15)

Reviewer 4 Report

The authors propose and evaluate a new transmission method for two-class sensed data in secure smart IoT systems.

The submission comes timely and the research performed is well motivated.

Solutions are technically solid

Presentation comes handy and it is easy to understand the content.

Related work is also fine giving a hint of state-of-the-art and how this proposal advances the field

Round 2

Reviewer 2 Report

The authors have well addressed my comments.

Reviewer 3 Report

After addressing the comments and remarks, the article is now in a better condition, suitable for publication. I appreciate the authors' detailed responses.